# Neural Arithmetic Logic Units

**Andrew Trask**[†‡]    **Felix Hill**[†]    **Scott Reed**[†]    **Jack Rae**[†♭]
**Chris Dyer**[†]    **Phil Blunsom**[†‡]
[†]DeepMind    [‡]University of Oxford    [♭]University College London
{atrask,felixhill,reedscot,jwrae,cdyer,pblunsom}@google.com

## Abstract

Neural networks can learn to represent and manipulate numerical information, but they seldom generalize well outside of the range of numerical values encountered during training. To encourage more systematic numerical extrapolation, we propose an architecture that represents numerical quantities as linear activations which are manipulated using primitive arithmetic operators, controlled by learned gates. We call this module a neural arithmetic logic unit (NALU), by analogy to the arithmetic logic unit in traditional processors. Experiments show that NALU-enhanced neural networks can learn to track time, perform arithmetic over images of numbers, translate numerical language into real-valued scalars, execute computer code, and count objects in images. In contrast to conventional architectures, we obtain substantially better generalization both inside and outside of the range of numerical values encountered during training, often extrapolating orders of magnitude beyond trained numerical ranges.

## 1  Introduction

The ability to represent and manipulate numerical quantities is apparent in the behavior of many species, from insects to mammals to humans, suggesting that basic quantitative reasoning is a general component of intelligence [5, 7].

While neural networks can successfully represent and manipulate numerical quantities given an appropriate learning signal, the behavior that they learn does not generally exhibit systematic generalization [6, 20]. Specifically, one frequently observes failures when quantities that lie outside the numerical range used during training are encountered at test time, even when the target function is simple (e.g., it depends only on aggregating counts or linear extrapolation). This failure pattern indicates that the learned behavior is better characterized by memorization than by systematic abstraction. Whether input distribution shifts that trigger extrapolation failures are of practical concern depends on the environments where the trained models will operate. However, considerable evidence exists showing that animals as simple as bees demonstrate systematic numerical extrapolation [? 7], suggesting that systematicity in reasoning about numerical quantities is ecologically advantageous.

In this paper, we develop a new module that can be used in conjunction with standard neural network architectures (e.g., LSTMs or convnets) but which is biased to learn systematic numerical computation. Our strategy is to represent numerical quantities as individual neurons without a nonlinearity. To these single-value neurons, we apply operators that are capable of representing simple functions (e.g., $+$, $-$, $\times$, etc.). These operators are controlled by parameters which determine the inputs and operations used to create each output. However, despite this combinatorial character, they are differentiable, making it possible to learn them with backpropagation [24].

We experiment across a variety of task domains (synthetic, image, text, and code), learning signals (supervised and reinforcement learning), and structures (feed-forward and recurrent). We find that our proposed model can learn functions over representations that capture the underlying numerical nature

of the data and generalize to numbers that are several orders of magnitude larger than those observed during training. We also observe that our module exhibits a superior numeracy bias relative to linear layers, even when no extrapolation is required. In one case our model exceeds a state-of-the-art image counting network by an error margin of 54%. Notably, the only modification we made over the previous state-of-the-art was the replacement of its last linear layer with our model.

## 1.1 Numerical Extrapolation Failures in Neural Networks

To illustrate the failure of systematicity in standard networks, we show the behavior of various MLPs trained to learn the scalar identity function, which is the most straightforward systematic relationship possible. The notion that neural networks struggle to learn identity relations is not new [14]. We show this because, even though many of the architectures evaluated below could theoretically represent the identity function, they typically fail to acquire it.

In Figure 1, we show the nature of this failure (experimental details and more detailed results in Appendix A). We train an autoencoder to take a scalar value as input (e.g., the number 3), encode the value within its hidden layers (distributed representations), then reconstruct the input value as a linear combination of the last hidden layer (3 again). Each autoencoder we train is identical in its parameterization (3 hidden layers of size 8), tuning (10,000 iterations, learning rate of 0.01, squared error loss), and initialization, differing only on the choice of nonlinearity on hidden layers. For each point in Figure 1, we train 100 models to encode numbers between $-5$ and 5 and average their ability to encode numbers between $-20$ and 20.

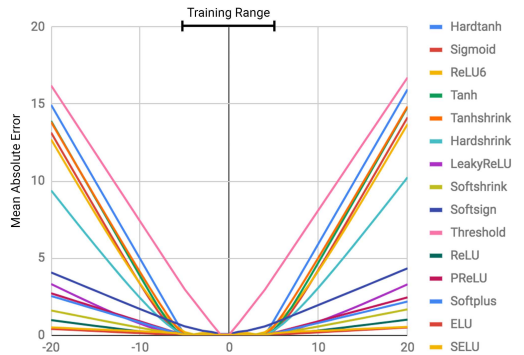

Figure 1: MLPs learn the identity function only for the range of values they are trained on. The mean error ramps up severely both below and above the range of numbers seen during training.

We see that even over a basic task using a simple architecture, all nonlinear functions fail to learn to represent numbers outside of the range seen during training. The severity of this failure directly corresponds to the degree of non-linearity within the chosen activation function. Some activations learn to be highly linear (such as PReLU) which reduces error somewhat, but sharply non-linear functions such as sigmoid and tanh fail consistently. Thus, despite the fact that neural networks are capable of representing functions that extrapolate, in practice we find that they fail to learn to do so.

## 2 The Neural Accumulator & Neural Arithmetic Logic Unit

Here we propose two models that are able to learn to represent and manipulate numbers in a systematic way. The first supports the ability to accumulate quantities additively, a desirable inductive bias for linear extrapolation. This model forms the basis for a second model, which supports multiplicative extrapolation. This model also illustrates how an inductive bias for arbitrary arithmetic functions can be effectively incorporated into an end-to-end model.

Our first model is the **neural accumulator** (NAC), which is a special case of a linear (affine) layer whose transformation matrix $\mathbf{W}$ consists just of $-1$'s, 0's, and 1's; that is, its outputs are additions or subtractions (rather than arbitrary rescalings) of rows in the input vector. This prevents the layer from changing the scale of the representations of the numbers when mapping the input to the output, meaning that they are consistent throughout the model, no matter how many operations are chained together. We improve the inductive bias of a simple linear layer by encouraging 0's, 1's, and $-1$'s within $\mathbf{W}$ in the following way.

Since a hard constraint enforcing that every element of $\mathbf{W}$ be one of $\{-1, 0, 1\}$ would make learning hard, we propose a continuous and differentiable parameterization of $\mathbf{W}$ in terms of unconstrained parameters: $\mathbf{W} = \tanh(\hat{\mathbf{W}}) \odot \sigma(\hat{\mathbf{M}})$ (where $\sigma$ corresponds to the sigmoid function). This form is convenient for learning with gradient descent and produces matrices whose elements are guaranteed

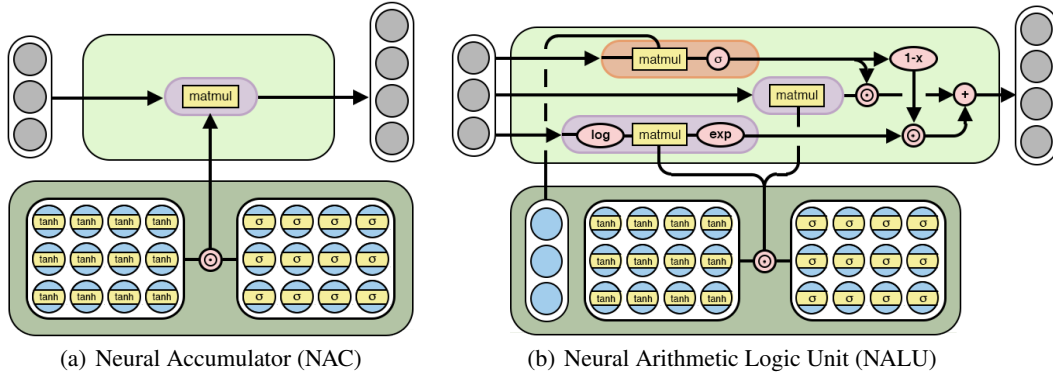

(a) Neural Accumulator (NAC)  (b) Neural Arithmetic Logic Unit (NALU)

Figure 2: The Neural Accumulator (NAC) is a linear transformation of its inputs. The transformation matrix is the elementwise product of $\tanh(\hat{\mathbf{W}})$ and $\sigma(\hat{\mathbf{M}})$. The Neural Arithmetic Logic Unit (NALU) uses two NACs with tied weights to enable addition/subtraction (smaller purple cell) and multiplication/division (larger purple cell), controlled by a gate (orange cell).

to be in $[-1, 1]$ and biased to be close to $-1$, $0$, or $1$.[1] The model contains no bias vector, and no squashing nonlinearity is applied to the output.

While addition and subtraction enable many useful systematic generalizations, a similarly robust ability to learn more complex mathematical functions, such as multiplication, may be be desirable. Figure 2 describes such a cell, the **neural arithmetic logic unit** (NALU), which learns a weighted sum between two subcells, one capable of addition and subtraction and the other capable of multiplication, division, and power functions such as $\sqrt{x}$. Importantly, the NALU demonstrates how the NAC can be extended with gate-controlled sub-operations, facilitating end-to-end learning of new classes of numerical functions. As with the NAC, there is the same bias against learning to rescale during the mapping from input to output.

The NALU consists of two NAC cells (the purple cells) interpolated by a learned sigmoidal gate $\mathbf{g}$ (the orange cell), such that if the add/subtract subcell's output value is applied with a weight of 1 (on), the multiply/divide subcell's is 0 (off) and vice versa. The first NAC (the smaller purple subcell) computes the accumulation vector $\mathbf{a}$, which stores results of the NALU's addition/subtraction operations; it is computed identically to the original NAC, (i.e., $\mathbf{a} = \mathbf{W}\mathbf{x}$). The second NAC (the larger purple subcell) operates in log space and is therefore capable of learning to multiply and divide, storing its results in $\mathbf{m}$:

$$
\begin{aligned}
\text{NAC:} \quad & \mathbf{a} = \mathbf{W}\mathbf{x} & & \mathbf{W} = \tanh(\hat{\mathbf{W}}) \odot \sigma(\hat{\mathbf{M}}) \\
\text{NALU:} \quad & \mathbf{y} = \mathbf{g} \odot \mathbf{a} + (1 - \mathbf{g}) \odot \mathbf{m} & & \mathbf{m} = \exp \mathbf{W}(\log(|\mathbf{x}| + \epsilon)), \ \mathbf{g} = \sigma(\mathbf{G}\mathbf{x})
\end{aligned}
$$

where $\epsilon$ prevents $\log 0$. Altogether, this cell can learn arithmetic functions consisting of multiplication, addition, subtraction, division, and power functions in a way that extrapolates to numbers outside of the range observed during training.

# 3  Related Work

Numerical reasoning is central to many problems in intelligence and by extension is an important topic in deep learning [5]. A widely studied task is counting objects in images [2, 4? , 25, 31, 33]. These models generally take one of two approaches: 1) using a deep neural network to segment individual instances of a particular object and explicitly counting them in a post-processing step or 2) learning end-to-end to predict object counts via a regression loss. Our work is more closely related to the second strategy.

Other work more explicitly attempts to model numerical representations and arithmetic functions within the context of learning to execute small snippets of code [32, 23]. Learning to count within a bounded range has also been included in various question-answer tasks, notably the BaBI tasks [29],

and many models successfully learn to do so [1, 18, 12]. However, to our knowledge, no tasks of this kind explicitly require counting beyond the range observed during training.

One can also view our work as advocating a new context for linear activations within deep neural networks. This is related to recent architectural innovations such as ResNets [14], Highway Networks [26], and DenseNet [15], which also advocate for linear connections to reduce exploding/vanishing gradients and promote better learning bias. Such connections improved performance, albeit with additional computational overhead due to the increased depth of the resulting architectures.

Our work is also in line with a broader theme in machine learning which seeks to identify, in the form of behavior-governing equations, the underlying structure of systems that extrapolate well to unseen parts of the space [3]. This is a strong trend in recent neural network literature concerning the systematic representation of concepts within recurrent memory, allowing for functions over these concepts to extrapolate to sequences longer than observed during training. The question of whether and how recurrent networks generalize to sequences longer than they encountered in training has been of enduring interest, especially since well-formed sentences in human languages are apparently unbounded in length, but are learned from a limited sample [9, 19, 28]. Recent work has also focused on augmenting LSTMs with systematic external memory modules, allowing them to generalize operations such as sorting [30, 11, 13], again with special interest in generalization to sequences longer than observed during training through systematic abstraction.

Finally, the cognitive and neural bases of numerical reasoning in humans and animals have been intensively studied; for a popular overview see Dehaene [5]. Our models are reminiscent of theories which posit that magnitudes are represented as continuous quantities manipulated by accumulation operations [? ], and, in particular, our single-neuron representation of number recalls Gelman and Gallistel's posited "numerons"—individual neurons that represent numbers [8]. However, across many species, continuous quantities appear to be represented using an approximate representation where acuity decreases with magnitude [22], quite different from our model's constant precision.

# 4   Experiments

The experiments in this paper test numeric reasoning and extrapolation in a variety of settings. We study the explicit learning of simple arithmetic functions directly from numerical input, and indirectly from image data. We consider temporal domains: the translation of text to integer values, and the evaluation of computer programs containing conditional logic and arithmetic. These supervised tasks are supplemented with a reinforcement learning task which implicitly involves counting to keep track of time. We conclude with the previously-studied MNIST parity task where we obtain state-of-the-art prediction accuracy and provide an ablation study to understand which components of NALU provide the most benefit.

## 4.1   Simple Function Learning Tasks

In these initial synthetic experiments, we demonstrate the ability of NACs and NALUs to learn to select relevant inputs and apply different arithmetic functions to them, which are the key functions they are designed to be able to solve (below we will use these as components in more complex architectures). We have two task variants: one where the inputs are presented all at once as a single vector (the static tasks) and a second where inputs are presented sequentially over time (the recurrent tasks). Inputs are randomly generated, and for the target, two values ($a$ and $b$) are computed as a sum over regular parts of the input. An operation (e.g., $a \times b$) is then computed providing the training (or evaluation) target. The model is trained end-to-end by minimizing the squared loss, and evaluation looks at performance of the model on held-out values from within the training range (interpolation) or on values from outside of the training range (extrapolation). Experimental details and more detailed results are in Appendix B.

On the static task, for baseline models we compare the NAC and NALU to MLPs with a variety of standard nonlinearities as well as a linear model. We report the baseline with the best median held-out performance, which is the Relu6 activation [17]. Results using additional nonlinearities are also in Appendix B. For the recurrent task, we report the performance of an LSTM and the best performing RNN variant from among several common architectures, an RNN with ReLU activations (additional recurrent baselines also in Appendix B).

|  |  | Static Task (test) | | | | Recurrent Task (test) | | | |
|---|---|---|---|---|---|---|---|---|---|
|  |  | Relu6 | None | NAC | NALU | LSTM | ReLU | NAC | NALU |
| Interpolation | $a + b$ | 0.2 | **0.0** | **0.0** | **0.0** | **0.0** | **0.0** | **0.0** | **0.0** |
| | $a - b$ | **0.0** | **0.0** | **0.0** | **0.0** | **0.0** | **0.0** | **0.0** | **0.0** |
| | $a \times b$ | 3.2 | 20.9 | 21.4 | **0.0** | **0.0** | **0.0** | 1.5 | **0.0** |
| | $a/b$ | **4.2** | 35.0 | 37.1 | 5.3 | **0.0** | **0.0** | 1.2 | **0.0** |
| | $a^2$ | 0.7 | 4.3 | 22.4 | **0.0** | **0.0** | **0.0** | 2.3 | **0.0** |
| | $\sqrt{a}$ | 0.5 | 2.2 | 3.6 | **0.0** | **0.0** | **0.0** | 2.1 | **0.0** |
| Extrapolation | $a + b$ | 42.6 | **0.0** | **0.0** | **0.0** | 96.1 | 85.5 | **0.0** | **0.0** |
| | $a - b$ | 29.0 | **0.0** | **0.0** | **0.0** | 97.0 | 70.9 | **0.0** | **0.0** |
| | $a \times b$ | 10.1 | 29.5 | 33.3 | **0.0** | 98.2 | 97.9 | 88.4 | **0.0** |
| | $a/b$ | 37.2 | 52.3 | 61.3 | 0.7 | **95.6** | 863.5 | >999 | >999 |
| | $a^2$ | 47.0 | 25.1 | 53.3 | **0.0** | 98.0 | 98.0 | 123.7 | **0.0** |
| | $\sqrt{a}$ | 10.3 | 20.0 | 16.4 | **0.0** | 95.8 | 34.1 | >999 | **0.0** |

Table 1: Interpolation and extrapolation error rates for static and recurrent tasks. Scores are scaled relative to a randomly initialized model for each task such that 100.0 is equivalent to random, 0.0 is perfect accuracy, and >100 is worse than a randomly initialized model. Raw scores in Appendix B.

Table 1 summarizes results and shows that while several standard architectures succeed at these tasks in the interpolation case, none of them succeed at extrapolation. However, in both interpolation and extrapolation, the NAC succeeds at modeling addition and subtraction, whereas the more flexible NALU succeeds at multiplicative operations as well (except for division in the recurrent task[2]).

## 4.2   MNIST Counting and Arithmetic Tasks

In the previous synthetic task, both inputs and outputs were provided in a generalization-ready representation (as floating point numbers), and only the internal operations and representations had to be learned in a way that generalized. In this experiment, we discover whether backpropagation can learn the representation of non-numeric inputs to NACs/NALUs.

In these tasks, a recurrent network is fed a series of 10 randomly chosen MNIST digits and at the end of the series it must output a numerical value about the series it observed.[3] In the MNIST Digit Counting task, the model must learn to count how many images of each type it has seen (a 10-way regression), and in the MNIST Digit Addition task, it must learn to compute the sum of the digits it observed (a linear regression). Each training series is formed using images from the MNIST digit training set, and each testing series from the MNIST test set. Evaluation occurs over held-out sequences of length 10 (interpolation), and two extrapolation lengths: 100 and 1000. Although no direct supervision of the convnet is provided, we estimate how well it has learned to distinguish digits by passing in a test sequences of length 1 (also from the MNIST test dataset) and estimating the accuracy based on the count/sum. Parameters are initialized randomly and trained by backpropagating the mean squared error against the target count vector or the sum.

Table 2 shows the results for both tasks. As we saw before, standard architectures succeed on held-out sequences in the interpolation length, but they completely fail at extrapolation. Notably, the RNN-tanh and RNN-ReLU models also fail to learn to interpolate to shorter sequences than seen during training. However, the NAC and NALU both extrapolate and interpolate well.

## 4.3   Language to Number Translation Tasks

Neural networks have also been quite successful in working with natural language inputs, and LSTM-based models are state-of-the-art in many tasks [10, 27, 16]. However, much like other numerical input, it is not clear whether representations of number words are learned in a systematic way. To test this, we created a new translation task which translates a text number expression (e.g., five hundred and fifteen) into a scalar representation (515).

| | MNIST Digit Counting Test | | | | MNIST Digit Addition Test | | | |
|---|---|---|---|---|---|---|---|---|
| | Classification | Mean Absolute Error | | | Classification | Mean Absolute Error | | |
| Seq Len | 1 | 10 | 100 | 1000 | 1 | 10 | 100 | 1000 |
| LSTM | 98.29% | 0.79 | 18.2 | 198.5 | 0.0% | 14.8 | 800.8 | 8811.6 |
| GRU | 99.02% | 0.73 | 18.0 | 198.3 | 0.0% | 1.75 | 771.4 | 8775.2 |
| RNN-tanh | 38.91% | 1.49 | 18.4 | 198.7 | 0.0% | 2.98 | 20.4 | 200.7 |
| RNN-ReLU | 9.80% | 0.66 | 39.8 | $1.4e10$ | 88.18% | 19.1 | 182.1 | 1171.0 |
| NAC | **99.23%** | **0.12** | **0.76** | **3.32** | **97.6%** | **1.42** | **7.88** | **57.3** |
| NALU | 97.6% | 0.17 | 0.93 | 4.18 | 77.7% | 5.11 | 26.8 | 248.9 |

Table 2: Accuracy of the MNIST Counting & Addition tasks for series of length 1, 10, 100, and 1000.

We trained and tested using numbers from 0 to 1000. The training set consists of the numbers 0–19 in addition to a random sample from the rest of the interval, adjusted to make sure that each unique token is present at least once in the training set. There are 169 examples in the training set, 200 in validation, and 631 in the test test. All networks trained on this dataset start with a token embedding layer, followed by encoding through an LSTM, and then a linear layer, NAC, or NALU.

| Model | Train MAE | Validation MAE | Test MAE |
|---|---|---|---|
| LSTM | 0.003 | 29.9 | 29.4 |
| LSTM + NAC | 80.0 | 114.1 | 114.3 |
| LSTM + NALU | 0.12 | **0.39** | **0.41** |

Table 3: Mean absolute error (MAE) comparison on translating number strings to scalars. LSTM + NAC/NALU means a single LSTM layer followed by NAC or NALU, respectively.

We observed that both baseline LSTM variants overfit severely to the 169 training set numbers and generalize poorly. The LSTM + NAC performs poorly on both training and test sets. The LSTM + NALU achieves the best generalization performance by a wide margin, suggesting that the multiplier is important for this task.

We show in Figure 3 the intermediate states of the NALU on randomly selected test examples. Without supervision, the model learns to track sensible estimates of the unknown number up to the current token. This allows the network to predict given tokens it has never seen before in isolation, such as e.g. `eighty`, since it saw

```
⌒  "three    hundred   and      thirty    four"
↳  3.05     299.9    301.3    330.1     334

⌒  "seven    hundred   and      two"
↳  6.98     699.9    701.3    702.2

⌒  "eighty   eight"
↳  79.6     88

⌒  "twenty   seven    and      eighty"
↳  18.2     27.0     29.1     106.1
```

Figure 3: Intermediate NALU predictions on previously unseen queries.

`eighty one`, `eighty four` and `eighty seven` during training.[4] The `and` token can be exploited to form addition expressions (see last example), even though these were not seen in training.

## 4.4 Program Evaluation

Evaluating a program requires the control of several logical and arithmetic operations and internal book-keeping of intermediate values. We consider the two program evaluation tasks defined in [32]. The first consists of simply adding two large integers, and the latter involves evaluating programs containing several operations (if statements, $+$, $-$). We focus on extrapolation: can the network learn a solution that generalizes to larger value ranges? We investigate this by training with two-digit input integers pulled uniformly from $[0, 100)$ and evaluating on random integers with three and four digits.

Following the setup of [32] we report the the percentage of matching digits between the rounded prediction from the model and target integer, however we handle numeric input differently. Instead of passing the integers character-by-character, we pass the full integer value at a single time step and

regress the output with an RMSE loss. Our model setup consists of a NALU that is "configured by" an LSTM, that is, its parameters $\hat{\mathbf{W}}$, $\hat{\mathbf{M}}$, and $\hat{\mathbf{G}}$ are learned functions of the LSTM output $\mathbf{h}_t$ at each timestep. Thus, the LSTM learns to control the NALU, dependent upon operations seen.

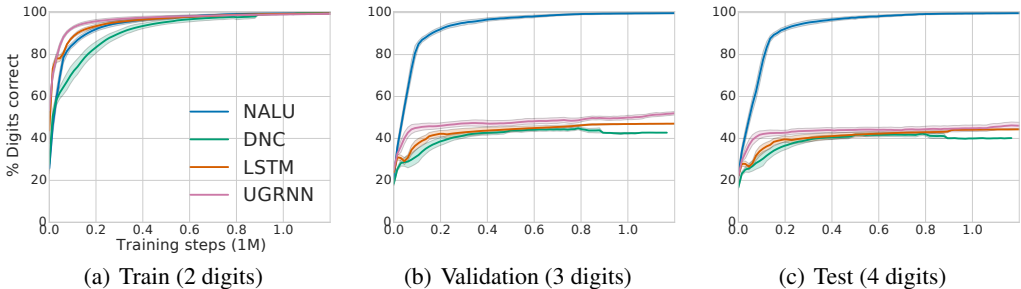

(a) Train (2 digits)    (b) Validation (3 digits)    (c) Test (4 digits)

Figure 4: Simple program evaluation with extrapolation to larger values. All models are averaged over 10 independent runs, $2\sigma$ confidence bands are displayed.

We compare to three popular RNNs (UGRNN, LSTM and DNC) and observe in both addition (Supplementary Figure 6) and program evaluation (Figure 4) that all models are able to solve the task at a fixed input domain, however only the NALU is able to extrapolate to larger numbers. In this case we see extrapolation is stable even when the domain is increased by two orders of magnitude.

## 4.5 Learning to Track Time in a Grid-World Environment

In all experiments thus far, our models have been trained to make numeric predictions. However, as discussed in the introduction, systematic numeric computation appears to underlie a diverse range of (natural) intelligent behaviors. In this task, we test whether a NAC can be used "internally" by an RL-trained agent to develop more systematic generalization to quantitative changes in its environment. We developed a simple grid-world environment task in which an agent is given a time (specified as a real value) and receives a reward if is arrives at a particular location at (and not before) that time. As illustrated in Figure 5, each episode in this task begins ($t = 0$) with the agent and a single target red square randomly positioned in a $5 \times 5$ grid-world. At each timestep, the agent receives as input a $56 \times 56$ pixel representation of the state of the (entire) world, and must select a single discrete action from $\{\text{UP}, \text{DOWN}, \text{LEFT}, \text{RIGHT}, \text{PASS}\}$. At the start of the episode, the agent also receives a numeric instruction $T$, which communicates the exact time the agent must arrive at its destination.

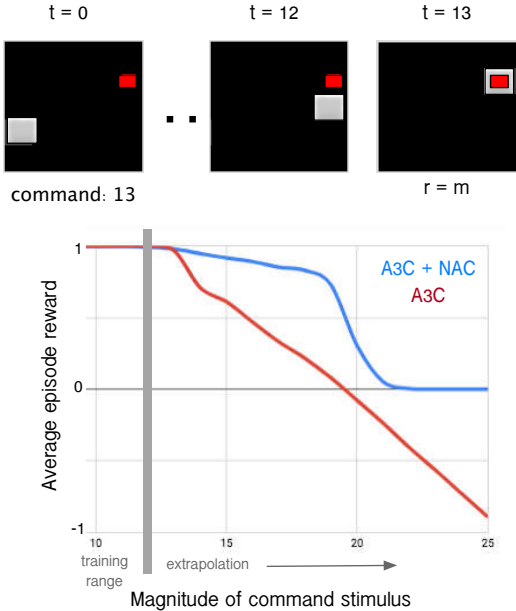

Figure 5: (above) Frames from the gridworld time tracking task. The agent (gray) must move to the destination (red) at a specified time. (below) NAC improves extrapolation ability learned by A3C agents for the dating task.

To achieve the maximum episode reward $m$, the agent must select actions and move around so as to first step onto the red square precisely when $t = T$. Training episodes end either when the agent reaches the red square or after timing out ($t = L$). We first trained a conventional A3C agent [21] with a recurrent (LSTM) core memory, modified so that the instruction $T$ was communicated to the agent via an additional input unit concatenated to the output of the agent's convnet visual module before being passed to the agent's LSTM core memory. We also trained a second variant of the same

architecture where the instruction was passed both directly to the LSTM memory and passed through an NAC and back into the LSTM. Both agents were trained on episodes where $T \sim \mathcal{U}\{5, 12\}$ (eight being the lowest value of $T$ such that reaching the target destination when $t = T$ is always possible). Both agents quickly learned to master the training episodes. However, as shown in Figure 5, the agent with the NAC performed well on the task for $T \leq 19$, whereas performance of the standard A3C agent deteriorated for $T > 13$.

It is instructive to also consider why both agents eventually fail. As would be predicted by consideration of extrapolation error observed in previous models, for stimuli greater than 12 the baseline agent behaves *as if the stimulus were still 12*, arriving at the destination at $t = 12$ (too early) and thus receiving incrementally less reward with larger stimuli. In contrast, for stimuli greater than 20, the agent with NAC never arrives at the destination. Note that in order to develop an agent that could plausibly follow both numerical and non-numeric (e.g. linguistic or iconic) instructions, the instruction stimulus was passed both directly to the agent's core LSTM and first through the NAC. We hypothesize that the more limited extrapolation (in terms of orders of magnitude) of the NAC here relative with other uses of the NAC was caused by the model still using the LSTM to encode numeracy to some degree.

### 4.6 MNIST Parity Prediction Task & Ablation Study

Thus far, we have emphasized the extrapolation successes; however, our results indicate that the NAC layer often performs extremely well at interpolation. In our final task, the MNIST parity task [25], we look explicitly at interpolation. Also, in this task, neither the input nor the output is directly provided as a number, but it implicitly invovles reasoning about numeric quantities. In these experiments, the NAC or its variants replace the last linear layer in the model proposed by Seguí et al. [25], where it connects output of the convnet to the prediction softmax layer. Since the original model had an affine layer here, and a NAC is a constrained affine layer, we look systematically at the importance of each constraint. Table 4 summarizes the performance of the variant models. As we see, removing the bias and applying nonlinearities to the weights significantly increases the accuracy of the end-to-end model, even though the majority of the parameters are not in the NAC itself. The NAC reduces the error of the previous best results by 54%.

| Layer Configuration | Test Acc. |
|---|---|
| Seguí et al. [25]: $\hat{\mathbf{W}}\mathbf{x} + \mathbf{b}$ | 85.1 |
| Ours: $\hat{\mathbf{W}}\mathbf{x} + \mathbf{b}$ | 88.1 |
| $\sigma(\hat{\mathbf{W}})\mathbf{x} + \mathbf{b}$ | 60.0 |
| $\tanh(\hat{\mathbf{W}})\mathbf{x} + \mathbf{b}$ | 87.6 |
| $\hat{\mathbf{W}}\mathbf{x} + \mathbf{0}$ | 91.4 |
| $\sigma(\hat{\mathbf{W}})\mathbf{x} + \mathbf{0}$ | 62.5 |
| $\tanh(\hat{\mathbf{W}})\mathbf{x} + \mathbf{0}$ | 88.7 |
| NAC: $(\tanh(\hat{\mathbf{W}}) \odot \sigma(\hat{\mathbf{M}}))\mathbf{x} + \mathbf{0}$ | **93.1** |

Table 4: An ablation study between an affine layer and a NAC on the MNIST parity task.

## 5 Conclusions

Current approaches to modeling numeracy in neural networks fall short because numerical representations fail to generalize outside of the range observed during training. We have shown how the NAC and NALU can be applied to rectify these two shortcomings across a wide variety of domains, facilitating both numerical representations and functions on numerical representations that generalize outside of the range observed during training. However, it is unlikely that NAC or NALU will be the perfect solution for every task. Rather, they exemplify a general design strategy for creating models that have biases intended for a target class of functions. This design strategy is enabled by the single-neuron number representation we propose, which allows arbitrary (differentiable) numerical functions to be added to the module and controlled via learned gates, as the NALU has exemplified between addition/subtraction and multiplication/division.

## 6 Acknowledgements

We thank Ed Grefenstette for suggesting the name Neural ALU and for valuable discussion involving numerical tasks. We also thank Steven Clark, whose presentation on neural counting work directly inspired this one. Finally, we also thank Karl Moritz Hermann, John Hale, Richard Evans, David Saxton, and Angeliki Lazaridou for valuable comments and discussion.

## Footnotes

[1]The stable points $\{-1, 0, 1\}$ correspond to the saturation points of either $\sigma$ or $\tanh$.

[2]Division is much more challenging to extrapolate. While our models limit numbers using nonlinearities, our models are still able to represent numbers that are very, very small. Division allows such small numbers to be in the denominator, greatly amplifying even small drifts in extrapolation ability.

[3]The input to the recurrent networks is the output the convnet in `https://github.com/pytorch/examples/tree/master/mnist`.

[4]Note slight accumulation for the word `and` owing to the spurious correlation between the use of the word `and` a subsequent increase in the target value.

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
