[Supplementary Material]

# A    Learning the Identity Function

| Name | Error | % Error | Graph |
|---|---|---|---|
| Hardtanh | 495.47 | 99.1 | |
| ReLU6 | 494.59 | 98.9 | |
| Softsign | 494.19 | 98.8 | |
| Tanh | 493.65 | 98.7 | |
| Sigmoid | 493.15 | 98.6 | |
| Threshold | 432.97 | 86.6 | |
| SELU | 365.66 | 73.1 | |
| ELU | 142.83 | 28.4 | |
| Softshrink | 119.17 | 23.8 | |
| ReLU | 83.63 | 16.7 | |
| LeakyReLU | 77.02 | 15.4 | |
| Tanhshrink | 57.03 | 11.4 | |
| Softplus | 37.60 | 7.5 | |
| PReLU | 14.66 | 2.9 | |
| None | <0.0001 | 0.0 | |

Table 5: Mean Absolute Reconstruction Error. 500 is equivalent to simply predicting 0 for every test datapoint. % Error is simply Error divided by 500. Scores represent averages over 100 models reconstructing all integer values from -1000 to 1000.

In Table 5, we show the average error results for each MLP trained on the identity function on the range from $-5$ to $5$ when evaluated on numbers ranging from $-1000$ to $1000$. Pictured on the right is a small graph demonstrating the plotted shape for each nonlinearity. The important thing to note is that nonlinearities which are sharply nonlinear exhibit greater extrapolation error than those which are only mildly nonlinear. An error score of 500 is equivalent to simply predicting 0 for each target prediction.

# B    Synthetic Arithmetic Tasks

In the static tasks, a vector $\mathbf{x} \in \mathbb{R}^{100}$ is given and the target is a scalar $y$ whose value is formed by first taking two random (but consistent) subsections and summing them $a = \sum_{i=m}^{n} x_i$ and $b = \sum_{j=p}^{q} x_j$. The target $y$ is then the result of applying different arithmetic functions to $a$ and $b$. The process must be learned end-to-end by each model.

This task is challenging because the test set comes in two forms. The first is the interpolation test set, which never proposes an example to the model requiring $a$, $b$, or $y$ to represent a number greater than exceeded during training. The extrapolation test set, however, always includes at least one value of $a$, $b$, or $y$ that is greater than observed during training in each example.

In order to test the recurrent variant of the NAC, we propose second set of tasks which are only a slight modification of the first. Instead of the first step requiring sums over an input vector $\mathbf{x} \in \mathbb{R}^{100}$, the sum is computed over a timeseries where each step $\mathbf{x_t} \in \mathbb{R}^{10}$, $a = \sum_{t}^{T} \sum_{i=m}^{n} x_{t,i}$, and $b = \sum_{t}^{T} \sum_{j=p}^{q} x_{t,j}$. Training and interpolation testing occur over sequences of length 10. Testing occurs over sequences of length 1000 (forcing $a$, $b$, and $y$ to take on values that are much larger than observed during training). All baseline experiments simply use an MLP with one hidden layer containing a non-linearity. For comparison, we stack two NALUs end-to-end. All models use a hidden layer size of 2, the minimally required size to solve the task.

In Table 7, we show the raw scores for MLPs trained with a wide variety of non-linearities over the hidden layer. As the scores for each task vary significantly, we also show scores for a randomly initialized model in the left column (headed "Random") for context. Table 8 normalizes all scores reported by dividing them by this "Random" column, making them more easily comparable.

| Tasks | Test | Random | Tanh | Sigmoid | Relu6 | Softsign | SELU | ELU | ReLU | Crelu | None | NAC | NALU |
|---|---|---|---|---|---|---|---|---|---|---|---|---|---|
| $a+b$ | I | 8.659 | .0213 | .0086 | .0144 | .0229 | .0031 | .0191 | .0040 | .0020 | .0017 | **<.0001** | **<.0001** |
|  | E | 120.9 | 52.37 | 51.14 | 51.56 | 47.78 | .0126 | .1064 | .0225 | .0127 | .0013 | **<.0001** | .0012 |
| $a-b$ | I | 6.478 | .0344 | .0183 | .0010 | .0616 | .0046 | .0510 | .0133 | .0035 | .0007 | **<.0001** | **.0001** |
|  | E | 42.14 | 11.54 | 9.627 | 12.22 | 9.734 | 5.60 | 3.458 | 6.108 | .0090 | .0007 | **<.0001** | **<.0001** |
| $a*b$ | I | 233.9 | 21.82 | 12.06 | 7.579 | 20.07 | 11.99 | 5.928 | 6.084 | 36.28 | 48.91 | 50.02 | **<.0001** |
|  | E | 3647 | 971.3 | 608.9 | 366.6 | 998.6 | 380.7 | 187.8 | 183.0 | 197.9 | 1076 | 1215 | **<.0001** |
| $\frac{a}{b}$ | I | 1.175 | .0514 | **.0421** | .0499 | .0558 | .1218 | .1233 | .1233 | .1011 | .4113 | .4363 | .0625 |
|  | E | 41.56 | 14.65 | 13.34 | 15.46 | 13.55 | 7.219 | 7.286 | 7.291 | 6.746 | 21.75 | 25.48 | **.2920** |
| $a^2$ | I | 151.7 | .3327 | .0672 | 1.027 | 1.479 | 6.462 | .1744 | .7829 | .1593 | 6.495 | 34.09 | **<.0001** |
|  | E | 4674 | 2168 | 2142 | 2195 | 1938 | 1237 | 406.6 | 569.4 | 408.4 | 1172 | 2490 | **<.0001** |
| $\sqrt{a}$ | I | 1.476 | .0124 | .0026 | .0080 | .0101 | .0046 | .0055 | .0265 | .0027 | .0329 | .0538 | **<.0001** |
|  | E | 4.180 | .4702 | .3571 | .4294 | .3237 | .0739 | .1006 | .5066 | .0622 | .8356 | .6876 | **<.0001** |

Table 6: Raw Mean Squared Error for all arithmetic tasks across activation functions. I/E refers to interpolation/extrapolation test sets respectively. Losses on the left refer to the decoder function where $a$ is the sum (a scalar) over one random subset of the input matrix and $b$ is the sum (a scalar) over another subset. The model must correctly predict the output of the function for each grid.

| Tasks | Relu6 | Softsign | Tanh | Sigmoid | SELU | ELU | ReLU | Crelu | None | NAC | NALU |
|---|---|---|---|---|---|---|---|---|---|---|---|
| | Interpolation Test Error - Relative to Random Initialization Baseline | | | | | | | | | | |
| $a+b$ | 0.2 | 0.3 | 0.2 | 0.1 | **0.0** | 0.2 | **0.0** | **0.0** | **0.0** | **0.0** | **0.0** |
| $a-b$ | **0.0** | 1.0 | 0.5 | 0.3 | 0.1 | 0.8 | 0.2 | 0.1 | **0.0** | **0.0** | **0.0** |
| $a \times b$ | 3.2 | 8.6 | 9.3 | 5.2 | 5.1 | 2.5 | 2.6 | 15.5 | 20.9 | 21.4 | **0.0** |
| $a/b$ | 4.2 | 4.7 | 4.4 | **3.58** | 10.4 | 10.5 | 10.5 | 8.6 | 35.0 | 37.1 | 5.3 |
| $a^2$ | 0.7 | 1.0 | 0.2 | **0.0** | 4.3 | 0.1 | 0.5 | 0.1 | 4.3 | 22.4 | **0.0** |
| $\sqrt{a}$ | 0.5 | 0.7 | 31.6 | 24.2 | 0.3 | 0.4 | 1.8 | 0.2 | 2.2 | 3.6 | **0.0** |
| | Extrapolation Test Error - Relative to Random Initialization Baseline | | | | | | | | | | |
| $a+b$ | 42.6 | 39.5 | 43.3 | 42.3 | **0.0** | **0.0** | **0.0** | **0.0** | **0.0** | **0.0** | **0.0** |
| $a-b$ | 29.0 | 23.1 | 27.3 | 22.8 | 13.3 | 8.2 | 14.5 | **0.0** | **0.0** | **0.0** | **0.0** |
| $a \times b$ | 10.1 | 27.4 | 26.6 | 16.7 | 10.4 | 5.1 | 5.0 | 5.4 | 29.5 | 33.3 | **0.0** |
| $a/b$ | 37.2 | 32.6 | 35.3 | 32.1 | 17.4 | 17.5 | 17.5 | 16.2 | 52.3 | 61.3 | 0.7 |
| $a^2$ | 47.0 | 41.5 | 46.4 | 45.8 | 26.5 | 8.7 | 12.2 | 8.7 | 25.1 | 53.3 | **0.0** |
| $\sqrt{a}$ | 10.3 | 7.7 | 11.2 | 8.5 | 1.7 | 2.4 | 12.1 | 1.5 | 20.0 | 16.4 | **0.0** |

Table 7: Static (non-recurrent) arithmetic error rates. Lower is better. Best models in bold. Scores relative to one randomly initialized model for each task. 100.0 is equivalent to random. 0.0 is perfect accuracy. Raw scores are in the Appendix.

| Model | T/I/E | $a$ | $a+b$ | $a-b$ | $a*b$ | $\frac{a}{b}$ | $a^2$ | $\sqrt{a}$ |
|---|---|---|---|---|---|---|---|---|
| Random | I | 38.03 | 30.86 | 30.17 | 30.69 | 29.12 | 30.92 | 28.84 |
| | E | 330341 | 336847 | 336769 | 336829 | 336451 | 336876 | 336452 |
| LSTM | T | .001771 | .004392 | .010654 | .017190 | .017723 | .014044 | .006919 |
| | I | .000544 | .037351 | .015579 | .004282 | .014142 | .016741 | .002258 |
| | E | 330097 | 323877 | 326746 | 330618 | 321622 | 330264 | 322305 |
| GRU | T | .002800 | .005177 | .011987 | .020600 | .024754 | .018683 | .006823 |
| | I | .000584 | .006819 | .026034 | .005473 | .059921 | .029642 | .002185 |
| | E | 330309 | 324036 | 321989 | 333336 | 318537 | 334332 | 321107 |
| RNN - TANH | T | .049635 | .049862 | .072149 | .161817 | .117094 | .076251 | .071165 |
| | I | .041697 | .020395 | .040893 | .169256 | .115955 | .340204 | .068180 |
| | E | 332760 | 324180 | 324000 | 330137 | 323908 | 329339 | 320840 |
| RNN - RELU | T | .058062 | .037226 | .044367 | .127584 | .096943 | .051219 | .045705 |
| | I | .005471 | .010728 | .019738 | .091305 | .073977 | .017803 | .007818 |
| | E | 325690 | 287928 | 238771 | 329636 | **2902047** | 330305 | 114725 |
| Neural Accumulator | T | .000002 | .000001 | .00002 | .46505 | .34161 | .69882 | .613314 |
| | I | .00020 | <.00001 | .000001 | .46862 | 0.350227 | .70909 | .617613 |
| | E | 1.8946 | .00004 | 0.01537 | 297800 | 3083319119 | 416701 | 2152274742 |
| Neural ALU | T | <.000001 | <.000001 | .000171 | .000074 | .001740 | .000164 | .000703 |
| | I | **<.000001** | **<.000001** | **.000001** | **.000003** | **.000431** | **.000001** | **.003492** |
| | E | **<.000001** | **.013131** | **.013843** | **47.0244** | >999999 | **2804.85** | **1671.81** |

Table 8: Mean Squared Error Loss values for all recurrent arithmetic tasks across baseline and proposed models. T/I/E refers to final training loss, interpolation loss, and extrapolation loss respectively. Best scores in bold.

| Model | $a$ | $a+b$ | $a-b$ | $a*b$ | $a/b$ | $a^2$ | $\sqrt{a}$ |
|---|---|---|---|---|---|---|---|
| | | | | Interpolation Test | | | |
| LSTM | 0.0 | 0.0 | 0.0 | 0.0 | 0.0 | 0.0 | 0.0 |
| GRU | 0.0 | 0.0 | 0.0 | 0.0 | 0.0 | 0.0 | 0.0 |
| tanh | 0.0 | 0.0 | 0.0 | 0.0 | 0.0 | 0.0 | 0.0 |
| ReLU | 0.0 | 0.0 | 0.0 | 0.0 | 0.0 | 0.0 | 0.0 |
| NAC | 0.0 | 0.0 | 0.0 | 1.5 | 1.2 | 2.3 | 2.1 |
| NALU | 0.0 | 0.0 | 0.0 | 0.0 | 0.0 | 0.0 | 0.0 |
| | | | | Extrapolation Test | | | |
| LSTM | 100.0 | 96.1 | 97.0 | 98.2 | 95.6 | 98.0 | 95.8 |
| GRU | 100.0 | 96.2 | 95.6 | 99.0 | **94.7** | 99.2 | 95.4 |
| tanh | 100.0 | 96.2 | 96.2 | 98.0 | 96.3 | 97.8 | 95.4 |
| ReLU | 98.6 | 85.5 | 70.9 | 97.9 | 862.5 | 98.0 | 34.1 |
| NAC | 0.0 | 0.0 | 0.0 | 88.4 | >999.9 | 123.7 | >999.9 |
| NALU | **0.0** | **0.0** | **0.0** | **0.0** | >999.9 | **0.0** | **0.0** |

Table 9: Recurrent use of NALU and NAC compared to modern recurrent architectures, evaluated using Mean Squared Error relative to a randomly initialized LSTM. 100.0 is equivalent to random. >100.0 is worse than random. 0 is perfect accuracy. Raw scores in appendix.

In Table 7, we observe that with very strong supervision over a simple task, most nonlinearities cannot learn functions requiring numeracy that generalize outside of the range observed during training. Note the consistency in these results with those in Table 5; common non-linearities with a very small output range catastrophically fail to extrapolate in any case, even with strong supervision, notably including Sigmoid and Tanh which are ubiquitous in recurrent neural networks, leading to results in Table 9.

In Table 9, we observe the accuracy of various baselines and our models when used recurrently. The tanh and relu in Table 9 refer to vanilla RNNs with tanh and relu respectively applied to hidden states. All models successfully learn to interpolate over learned functions of the input. However, the when attempting to extrapolate a function learned over series of length 10 to series of length 1000, the NAC and NALU significantly outperform the baselines. Division, however, was quite challenging to extrapolate and no models were able to solve the task in a way that extrapolates. Baselines generally predicted a fixed range. However, the NAC and NALU underestimated the denominator, leading to quite significant error during extrapolation.

While these simple experiments are quite numerous, they exist to make clear a simple idea. Across a wide variety of nonlinearities and recurrent combinations of nonlinearities, most architectures can fit a training dataset requiring arithmetic; many can even learn to generalize to a test set if the numbers are within the same bounded range. However, modern neural architectures are ill-equipped to learn arithmetic in a systematic way. We did observe a few narrow exceptions in Table 7, but we will later show that even these do not hold when the supervised signal is more realistic (such as when this model is merely a piece of a large end-to-end architecture). It is the systematic numerical representations present in NAC and NALU that create the desirable learning bias leading to accurate extrapolation. This will continue to be the key to systematic extrapolation in future experiments as well.

## C  MNIST Counting

We report the performance of MNIST counting and addition for the full set of models considered in Table 10.

| | MNIST Digit Counting (test) | | | | MNIST Digit Addition (test) | | | |
|---|---|---|---|---|---|---|---|---|
| | Classification | Mean Squared Error | | | Classification | Mean Squared Error | | |
| Seq Len | 1 | 10 | 100 | 1000 | 1 | 10 | 100 | 1000 |
| LSTM | 98.29% | 1.14 | 181.06 | 19883 | 0.0% | 168.18 | 321738 | 38761851 |
| GRU | 99.02% | 1.12 | 180.95 | 19886 | 0.0% | 168.09 | 321826 | 38784947 |
| RNN-tanh | 38.91% | 1.53 | 226.95 | 20346 | 0.0% | 167.19 | 321841 | 38784910 |
| RNN-ReLU | 9.80% | 0.54 | 160.80 | 19608 | 88.18% | 4.29 | 882.87 | 10969180 |
| NAC | **99.23%** | **0.03** | **0.26** | **3** | **97.58%** | **2.82** | **28.11** | **280.89** |
| NALU | 97.62% | 0.08 | 0.90 | 17 | 77.73% | 18.22 | 1199.12 | 114303 |

Table 10: Accuracy of the MNIST Counting & Addition tasks for series of length 1, 10, 100, and 1000.

## D  Language To Number Translation Tasks

For the LSTM, we tried both summing the preceding states as output instead of using the final state and found this to be slightly advantageous to simply outputting the final state, as shown in Table 11. This *summed state* LSTM is what we use for comparison with the NAC and NALU in table 3. We train all models for 300K steps of gradient descent on the whole training set using Adam.

We selected the best model by validation loss over layer sizes $\{16, 32\}$, learning rates $\{0.01, 0.001\}$, and 10 initializations.

| Model | Train MAE | Validation MAE | Test MAE |
|---|---|---|---|
| LSTM w/ final state | 0.0085 | 32.1 | 32.2 |
| LSTM w/ summed states | 0.003 | 29.9 | 29.4 |

Table 11: Mean absolute error (MAE) comparison on translating number strings to scalars with LSTM state aggregation methods. Summing states improved generalization slightly, but

## E  Program Evaluation

The addition task, which is the simpler variant of the two program evaluation tasks considered, is simple for all models to solve. However all models fail to generalize except for the recurrent NALU. The UGRNN proves much better than the LSTM and DNC, likely due to the simple linear update of the state.

(a) Train (2 digits)  (b) Validation (3 digits)  (c) Test (4 digits)

Figure 6: Summing a sequence of two random integers with extrapolation to larger values. All models are averaged over 10 independent runs, $2\sigma$ confidence bands are displayed.

# F    NAC/NALU Using Imaginary Activations

One limitation in the aforementioned implementation is that it cannot learn to multiply negative numbers. Gabor Gulyas (guyko81@gmail.com) proposes that we could solve this issue by forward propagating with imaginary numbers. Thus, the proposed NAC and NALU formulas become:

$$\text{NAC:} \quad \mathbf{a} = \mathbf{Wx} \qquad\qquad \mathbf{W} = \tanh(\hat{\mathbf{W}}) \odot \sigma(\hat{\mathbf{M}})$$
$$\text{NALU:} \quad \mathbf{y} = \mathbf{g} \odot \mathbf{a} + (1 - \mathbf{g}) \odot \mathbf{m} \qquad \mathbf{m} = \exp \mathbf{W}(\log(\mathbf{x} + \epsilon)), \ \mathbf{g} = \sigma(\mathbf{Gx})$$

where all activations and gradients are computed including an imaginary term. We have evaluated this task on the Static Arithmetic tasks and found that we get nearly identical performance. However, we still need to mask out zero activations (by adding $\epsilon$ to them) because (in log space) they correspond to an infinite value, causing numerical stability issues.

| Tasks | Test | Random | Tanh | Sigmoid | Relu6 | Softsign | SELU | ELU | ReLU | Crelu | None | NAC | NALU |
|---|---|---|---|---|---|---|---|---|---|---|---|---|---|
| $a + b$ | I | 8.659 | .0213 | .0086 | .0144 | .0229 | .0031 | .0191 | .0040 | .0020 | .0017 | **<.0001** | **.0001** |
| | E | 120.9 | 52.37 | 51.14 | 51.56 | 47.78 | .0126 | .1064 | .0225 | .0127 | .0013 | **<.0002** | .17 |
| $a - b$ | I | 6.478 | .0344 | .0183 | .0010 | .0616 | .0046 | .0510 | .0133 | .0035 | .0007 | **<.0001** | **<.0001** |
| | E | 42.14 | 11.54 | 9.627 | 12.22 | 9.734 | 5.60 | 3.458 | 6.108 | .0090 | .0007 | **<.0001** | **<.0001** |
| $a * b$ | I | 233.9 | 21.82 | 12.06 | 7.579 | 20.07 | 11.99 | 5.928 | 6.084 | 36.28 | 48.91 | 50.13 | **<.0001** |
| | E | 3647 | 971.3 | 608.9 | 366.6 | 998.6 | 380.7 | 187.8 | 183.0 | 197.9 | 1076 | 1217 | **<.0001** |
| $\frac{a}{b}$ | I | 1.175 | .0514 | **.0421** | .0499 | .0558 | .1218 | .1233 | .1233 | .1011 | .4113 | .4366 | .1659 |
| | E | 41.56 | 14.65 | 13.34 | 15.46 | 13.55 | 7.219 | 7.286 | 7.291 | 6.746 | 21.75 | 25.45 | **26.09** |
| $a^2$ | I | 151.7 | .3327 | .0672 | 1.027 | 1.479 | 6.462 | .1744 | .7829 | .1593 | 6.495 | 34.09 | **<.0001** |
| | E | 4674 | 2168 | 2142 | 2195 | 1938 | 1237 | 406.6 | 569.4 | 408.4 | 1172 | 2489 | **<.0001** |
| $\sqrt{a}$ | I | 1.476 | .0124 | .0026 | .0080 | .0101 | .0046 | .0055 | .0265 | .0027 | .0329 | .0538 | **<.0001** |
| | E | 4.180 | .4702 | .3571 | .4294 | .3237 | .0739 | .1006 | .5066 | .0622 | .6866 | .6876 | **<.0001** |

Table 12: Raw Mean Squared Error for all arithmetic tasks across activation functions. The NAC/NALU columns in this table were trained using imaginary activations and the formula specified above this table (here in Appendix F). I/E refers to interpolation/extrapolation test sets respectively. Losses on the left refer to the decoder function where $a$ is the sum (a scalar) over one random subset of the input matrix and $b$ is the sum (a scalar) over another subset. The model must correctly predict the output of the function for each grid.

Accuracy overall for this model is very similar to that of the one proposed above, although extrapolation for division appears to be less stable. Future work will explore further improving the numerical stability of this process in a wider variety of contexts.