[Reviews · NeurIPS 2018]

Reviewer 1



This paper presents novel arithmetic units called NAC and NALU for neural networks. The proposed NAC unit is used for calculating additions and subtractions of the input vector; NALU calculates multiplications by applying a logarithm before the NAC unit. The experimental results show that the NAC/NALU units worked well on numerical addition and counting tasks--even when the lengths of numbers in the test data are different than those of training data. This work is motivated by the problem of learning the identity function, Figure 1 shows that many activation functions do not work in this problem. What is the performance of NAC/NALU on learning the identity function? Would it be a straight line as expected? Figure 1 also shows that some activation functions perform well on this problem (although hard to identify on the figure because of the overlapping lines and colors). It would be better if the authors can show their performance on the same tasks in the experiment section. Another problem is that the illustration and caption of Figure 2 are not very clear, e.g., what is the three blue dots on the left of the tanh matrix? How do the gates work? Why are they designed like this? Please provide a more detailed description of the NALU unit. The experiment on reinforcement learning shows that although NAC did not work on extrapolation tasks as expected, it degenerates slower than the compared method. How about NALU's performance on this task? The reward of A3C can lower than 0, why A3C+NAC did not get lower than 0? Some other questions: 1. What is the \hat{M} in line 87? 2. I would prefer to see NAC/NALU applied to more complex tasks, such as the object counting tasks introduced in section 1.

Reviewer 2



The paper proposes a method to extrapolate numbers outside the training range in neural networks. It proposes two models in order to solve this task. The neural accumulator that transforms a linear layer to output additions or subtractions and the neural arithmethic logic unit which can learn more complex mathematical functions. The tackled problem of extrapolation is an interesting one as neural networks generally perform not so well in such tasks. The paper itself is well motivated in the introduction and the example which compares different existing structures against each other adds additional clarity that this task should be studied. The method is interesting, novel and sound. It is well presented through good explanations and the diagrams in figure 2. The large experiment section further supports the proposed approach. It is compared in different problem settings against many different baselines and even against each other. The promise as well as the limitations are discussed. The presentation of the idea overall was well done. However, it does have some gaps. A short background section, for example, could have introduced the reader to extrapolation as a problem and could have reiterated the most important neural network mechanisms as later a deeper knowledge is expected. Furthermore, the mathematical notation (M) used in line 58 onwards is never formally introduced. Towards the end the authors mention that their work could also be seen as a different principled approach of designing neural networks in that it biases towards certain outcomes. the paper could have discussed this in more detail as this seems wider applicable than to only extrapolation problems and, therefore, more relevant to more readers. Overall, the paper presented its main contributions clearly and showed in a large result section the importance as well as the success of their method. While the more general underlying principle would have been interesting to explore and a background section might have added additional clarity the paper's novel ideas and impressive results still outline a significant contribution.

Reviewer 3



The paper proposes neural arithmetic logical units (NALUs) for learning numerical related tasks with deep neural networks. Specifically, NALUs explicitly enforce learning the systematic numerical operations, so the networks can perform well on values not seen in the training set. The proposed method is clean, general and easily applicable to existing neural networks. The paper gives strong empirical evidence on various numerical tasks and shows superior performance on networks with NALUs. I think the paper might need more studies like Figure 3 on both NALU and ordinary networks to better understand the network behaviors on various tasks. Update: Thanks for the rebuttal! I have read all reviewers' reviews and the corresponding response. There are some flaws (mathematical notions, figure clarity, and lack of background work discussion), but overall I think the paper is of good quality.